# Russian–German Astroparticle Data Life Cycle Initiative

Igor Bychkov [1,2], Andrey Demichev [3], Julia Dubenskaya [3], Oleg Fedorov [4], Andreas Haungs [5], Andreas Heiss [6], Donghwa Kang [5], Yulia Kazarina [4], Elena Korosteleva [3], Dmitriy Kostunin [5,*], Alexander Kryukov [3], Andrey Mikhailov [1], Minh-Duc Nguyen [3], Stanislav Polyakov [3], Evgeny Postnikov [3], Alexey Shigarov [1,2], Dmitry Shipilov [4], Achim Streit [6], Victoria Tokareva [5], Doris Wochele [5], Jürgen Wochele [5] and Dmitry Zhurov [4]

[1]  Matrosov Institute for System Dynamics and Control Theory, Siberian Branch of Russian Academy of Sciences, Lermontov st. 134, P. O. Box 292, 664033 Irkutsk, Russia; bychkov@icc.ru (I.B.); mikhailov@icc.ru (A.M.); shigarov@icc.ru (A.S.)

[2]  Institute Of Mathematics, Economics and Informatics, Irkutsk State University, Gagarin Blvd. 20, Irkutsk 664003, Russia

[3]  Skobeltsyn Institute of Nuclear Physics, Lomonosov Moscow State University, Leninskiye Gory 1(2), 119991 Moscow, Russia; demichev@theory.sinp.msu.ru (A.D.); jdubenskaya@gmail.com (J.D.); elkrs@yandex.ru (E.K.); kryukov@theory.sinp.msu.ru (A.K.); conqueror@dec1.sinp.msu.ru (M.-D.N.); s.p.polyakov@gmail.com (S.P.); evgeny.post@gmail.com (E.P.)

[4]  Applied Physics Institute, Irkutsk State University, Gagarin Blvd. 20, 664003 Irkutsk, Russia; Offedoroff@yandex.ru (O.F.); lutien777@mail.ru (Y.K.); justforprince@gmail.com (D.S.); sidney28@ya.ru (D.Z.)

[5]  Institute for Nuclear Physics, Karlsruhe Institute of Technology, KIT, 76021 Karlsruhe, Germany; andreas.haungs@kit.edu (A.H.); donghwa.kang@kit.edu (D.K.); victoria.tokareva@kit.edu (V.T.); doris.wochele@kit.edu (D.W.); juergen.wochele@kit.edu (J.W.)

[6]  Steinbuch Centre for Computing, Karlsruhe Institute of Technology, KIT, 76021 Karlsruhe, Germany; andreas.heiss@kit.edu (A.H.); achim.streit@kit.edu (A.S.)

*  Correspondence: editor@astroparticle.online

**Abstract:** Modern large-scale astroparticle setups measure high-energy particles, gamma rays, neutrinos, radio waves, and the recently discovered gravitational waves. Ongoing and future experiments are located worldwide. The data acquired have different formats, storage concepts, and publication policies. Such differences are a crucial point in the era of Big Data and of multi-messenger analysis in astroparticle physics. We propose an open science web platform called ASTROPARTICLE.ONLINE which enables us to publish, store, search, select, and analyze astroparticle data. In the first stage of the project, the following components of a full data life cycle concept are under development: describing, storing, and reusing astroparticle data; software to perform multi-messenger analysis using deep learning; and outreach for students, post-graduate students, and others who are interested in astroparticle physics. Here we describe the concepts of the web platform and the first obtained results, including the meta data structure for astroparticle data, data analysis by using convolution neural networks, description of the binary data, and the outreach platform for those interested in astroparticle physics. The KASCADE-Grande and TAIGA cosmic-ray experiments were chosen as pilot examples.

**Keywords:** astroparticle physics; cosmic rays; data life cycle management; data curation; meta data; Big Data; deep learning; open data

## 1. Introduction

Research in astroparticle physics addresses some of the most fundamental questions in nature. Various experiments in astroparticle physics span almost the whole spectrum of cosmic rays and all types of radiation [1,2]. There is an intimate connection between measurements and theoretical descriptions of astrophysical phenomena to provide the foundation for the sophisticated models of macroscopic astrophysical systems. Scientists have to share their incredibly detailed observations obtained both by ground-based and space-based devices to study processes in astrophysical environments. Moreover, information from various messengers, like charged particles [3] or neutrons [4], gamma-rays [5,6], or neutrinos [7], measured by different large-scale facilities distributed across the globe, has to be combined to obtain increased knowledge of the high-energy Universe. While neutrinos and gamma-rays point directly to the source, cosmic rays are heavily declined in the galactic and extra-galactic magnetic fields. Nevertheless, their anisotropy can indicate the nearest sources [8] as well as point to the possible distant sources of ultra-high-energy cosmic rays [9]. Furthermore, with increasing cosmic ray energy, the impact of magnetic fields decreases, which allows one to perform extremely-high-energy proton astronomy.

Some of the messengers (i.e., neutral particles and gamma rays) can be directly traced back to their sources, whereas the others (i.e., charged particles), though not supporting a simple "point-and-shoot" type of analysis, can nonetheless bring a different but crucial piece of knowledge. For example, due to its high sensitivity, the CTA project [10] will be capable of detecting the first gamma-ray signature of the most powerful accelerators of ultra-high-energy cosmic rays [11]. Several attempts have been made to search for correlations between high-energy neutrinos and charged cosmic rays, as well as between gamma rays and cosmic rays [12–16]. This kind of investigation is much easier to perform using a common set of astrophysical data and convenient tools to work with it. Besides, combining signals from different messengers can improve data analysis quality and lower detection thresholds for individual facilities by careful coincidence analysis [17].

Contemporaneous multi-wavelength and multi-messenger studies are important to provide comprehensive coverage of cosmic sources. However, astronomy is already provided with dedicated research and data centers: for example, ESO allows conducting various combined research programs [18,19]. That is why we are currently concentrating on combining astroparticle measurement data. This approach, also called multi-messenger astroparticle physics, requires a diverse set of astrophysical data to be made public and accessible to anyone.

The current trend, not only in astroparticle physics but also in other research areas, is that consumers from all over the world can download scientific data or use them online as soon as they are published. It demonstrates the power of the Internet and the ability of the scientific community to share data instantly with colleagues and with the general public. Some experiments in astroparticle physics have already adopted this fascinating idea, and have included their scientific data in electronic publishing, such as at the KASCADE Cosmic Ray Data Centre (KCDC) [20]. KCDC, presently in its beta phase, is a web portal where the KASCADE (and its extension KASCADE-Grande) [21] scientific data are made available for the interested public. However, KCDC is a small project driven within the KASCADE-Grande experiment only. In Russia, there is the operating Tunka Advanced Instrument for cosmic rays and Gamma Astronomy (TAIGA) [22] facility, which continuously produces data. There are many scientific reasons to bring together the TAIGA and KASCADE-Grande data to perform combined analysis with sophisticated methodical approaches (e.g., deep learning). Currently, an infrastructure for combined analysis using data from different facilities, which might be the next step in Big Data analytics, is not available yet.

This paper presents the current status of the Russian–German astroparticle data life cycle initiative also referred to as ASTROPARTICLE.ONLINE. The initiative strives to develop an open science system to be able to collect, store, and analyze astrophysical data having the TAIGA and KASCADE-Grande experimental facilities as initial data providers. The project, ASTROPARTICLE.ONLINE, aims at a common data portal for two independent observatories and at the same time for the consolidation and

maturation of an analysis and data centre of astroparticle physics experiments. There are four main goals of the project:

1. KCDC extension: the already-existing data centre released an initial dataset of parameters of more than 400 million extensive air showers of the concluded KASCADE-Grande experiment. The initiative extends KCDC with more scientific data from the TAIGA experiment (i.e., current/up-to-date data), allowing on-the-fly multi-messenger-analysis. Our goal is to extend and improve KCDC and make it more attractive to a broader user community.

2. Big Data science software: such an extension of the data centre allowing not only access to the data but also the possibility of developing specific analysis methods and corresponding simulations in one environment requires a move to the most modern computing, storage, and data access concepts, which is only possible by a close co-operation between the participating groups from both physics and information technology. A possible concept to reach this goal is the installation of a dedicated so-called "data life cycle lab", which this project is aiming for. Dedicated access, storage, and interface software have to be developed.

3. Reliability tests: some specific analysis of the data provided by the new data centre will be performed to test the entire concept. The results will give important contributions and confidence in the project as a valuable scientific tool.

4. Go for the public: the full outreach aspect of the project, including sample applications for all user levels, from pupils to the directly involved scientists to theoreticians, with detailed tutorials and documentation is an important goal of the project.

The novelty of the proposed approach is reflected in developing integrated solutions for:

- Distributed data storage algorithms and techniques with a common metadata catalog to provide a common information space of the distributed repository;
- Data transmission algorithms as well as simultaneous data transmission from several data repositories, thus significantly reducing load time;
- Deep-learning techniques for identifying mass groups of impinging cosmic particles and their properties in a fully remote access mode;
- A KCDC-based prototype system of Big Data analysis and exporting the experimental data from KASCADE-Grande and TAIGA to test the technology of efficient data life cycle management.
- An educational system based on the HUBzero[1] platform dedicated to astroparticle physics.

This paper's contribution consists of the following results:

- We defined the concept of an astroparticle data life cycle, covering the following issues: storage, simulations, analysis, education, open access, and archive of astroparticle data (Section 2).
- We introduced a metadata architecture for cosmic ray experiments that aims at describing and searching for all events from KASCADE-Grande and TAIGA experiments in a centralized database (Section 3.1).
- We proposed and estimated a novel technique for particle identification in imaging air Cherenkov telescopes based on deep learning. The technique was implemented with two well-known deep learning platforms—PyTorch and TensorFlow (Section 3.2).
- We developed educational resources for teaching students in the field of astroparticle physics. The resources were implemented with HUBzero, an open-source software platform for building scientific collaboration websites (Section 3.3).
- We examined the applicability of some data format description languages for documenting, parsing, and verifying raw binary data produced in both experiments. The implemented formal

---

[1] https://hubzero.org.

specifications of all file formats allowed source code to be automatically generated for data reading libraries in target programming languages (Section 3.4).

## 2. Concept of an Astroparticle Data Life Cycle

At present, an exponential growth in the amount of experimental data can be observed. While there were 1–10 terabytes of data per year in astrophysics 10 to 15 years ago, new experimental facilities generate data sets ranging in size from 100 s to 1000 s of terabytes per year. For example, while the Integral satellite [23] downloaded 1.2 gigabytes of data to the ground per day in 2002, the Gaia spacecraft [24] now transfers about 5 gigabytes per day. Another example is the ground-based experiment LSST [25], which will provide more than 3 gigapixels per image every 15 s. It is expected to produce about 10 petabytes of information per year.

These trends give rise to a number of emerging issues in Big Data management. Obviously, various activities should be performed continuously across all stages of the data life cycle to support the data management effectively: collecting and storing data, processing and analyzing data, refining physical models, making preparations for publication, as well as reprocessing data. An important topic for modern science in general and astroparticle physics in particular is open science, the model of free access to scientific data (e.g., [26]): data are accessible not only to collaboration members but to all levels of an inquiring society, amateurs or professionals. This approach is especially important in the age of Big Data and Open Science culture, when deep analysis of the experimental data cannot be performed within a single collaboration.

Usually, basic research in the fields of particle physics, astroparticle physics, nuclear physics, astrophysics, or astronomy is performed in large international collaborations with particularly huge infrastructures producing a big volume of valuable scientific data. To efficiently use all the information to solve the still mysterious question about the origin of matter and the Universe, a broad, simple, and sustainable access to the scientific data from these infrastructures has to be provided.

In a general way, such a global data centre should provide a vast functionality, at least covering the following pillars (see Figure 1):

1.  Data availability: all participating researchers of the individual experiments or facilities need fast and simple access to the relevant data.
2.  Simulations and methods development: to prepare the analysis of the data the researchers need an environment with mighty computing power for the production of relevant simulations and the development of new methods (e.g., by deep machine learning).
3.  Analysis: fast access to the (probably distributed) Big Data from measurements and simulations is needed.
4.  Education in data science: the handling of the data centers as well as the processing of the data needs specialized education in "Big Data science".
5.  Open access: it is increasingly important to provide the scientific data not only to the internal research community but also to the interested public.
6.  Data archive: the valuable scientific data need to be preserved for later reuse.

Whereas in astronomy and particle physics data centers that fulfill a part of these requirements are already established (although not the same parts), only first attempts are presently under development in astroparticle physics. The reason is the diversity of the experimental facilities in astroparticle physics and their distribution all over the world (partly in very harsh environments), without dedicated research centers like CERN[2] in particle physics, FAIR[3] in nuclear physics, or ESO[4] in astronomy.

---

[2]    https://home.cern.
[3]    https://fair-center.eu.
[4]    https://eso.org.

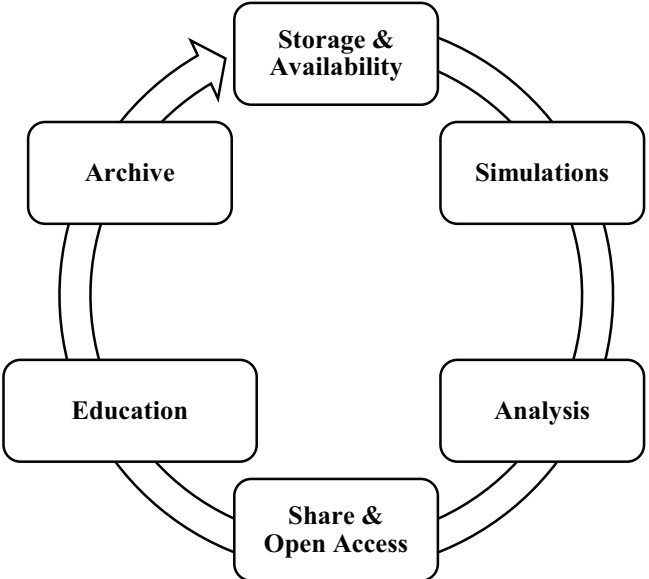

**Figure 1.** Concept of an astroparticle data life cycle.

Presently, astroparticle physicists do have access to data and several attempts have been made to provide it to their community in individual pillars. Namely, a small portion of the Tier computing centers are used by astroparticle physics experiments [27], for example, less than 5% of GridKa[5] by the Pierre Auger Collaboration [28]. In addition, first IceCube [29] or Pierre Auger data can be found in Astronomical Virtual Observatories like GAVO[6]. KCDC is an example of a first public release of scientific data. However, these attempts are uncoordinated and mostly specific to individual experiments or collaborations.

It is obvious that astroparticle physics has become a data-intensive science with many terabytes of data and often with tens of measured parameters associated with each observation. Moreover, new highly complex and massively large datasets are expected from novel and more complex scientific instruments, as well as simulated data requiring interpretation that will become available in the next decades, probably largely used by the community. Handling and exploring these new data volumes, and actually making unexpected scientific discoveries, pose a considerable technical challenge that requires the adoption of new approaches in using computing and storage resources and in organizing scientific collaborations. In addition, scientific education and science communication, where sophisticated public data centers will play a key role, are requested.

The methods for the successful performance of the initiative are a mixture of the work of computer scientists (i.e., sophisticated programming and the usage of modern tools to handle Big Data). This is a major focus of the project and must not be underestimated, as there is not yet a standard tool to do it. Typical challenges in Big Data processing include data searching, sharing, storage, transfer, and visualization. In information sciences, Big Data is related to the use of predictive analytics or certain other advanced methods to extract scientific value from data. We will apply the concepts of Big Data analytics to astroparticle physics (i.e., the understanding of detecting, reconstructing, and interpreting particles coming from the deep Universe to Earth). Finally, associated outreach tasks and the use of social media will be developed, as a distinct dissemination plan is required for the creation of a public data centre available to society as a whole.

---

5     http://www.gridka.de.
6     http://www.g-vo.org.

The FAIR Data Principles are the supreme guideline for all the research data management issues within this project. The FAIR principles[7] intend to provide guidelines for improving the usability of digital assets: Findable—the first step in (re-)using data is to find them. Accessible—once the user has found the required data, they must know how to be integrated with them, possibly including authentication and authorization. Interoperable—the data usually have to be integrated with other data. Reusable—the goal is to optimize the reuse of the data and allow the mining of archived data. In order to achieve this, metadata and data should be well-described so that they can be used in different ways.

### 2.1. Storage and Availability

A major goal of the project is to allow scientists to search for necessary data using specific requests. A request is a set of conditions and logical operations on them which define what kind of data a user wants to obtain. All requests are processed by special metadata servers using only the metadata information. Context search within data files will not be available. If one needs to carry out more sophisticated requests, the appropriate information must be extracted from the data and inserted into the meta registry.

The data itself are stored in some local data storage which collects raw data from astroparticle facilities such as TAIGA, KASCADE-Grande, etc. Each storage has its own data storage format, directory structure, and policy. We do not intend to touch the internal structure of the storage and traditions of the physics community. Therefore, to provide access to the data storage, it is necessary to deploy a RESTful service which will unify the external interface of all storage instances. We call such a service an adapter.

One of the possible solutions to implement such an adapter is CERNVM-FS[8]. CERNVM-FS exports a local file system over the Internet in read-only mode. Read-only data access is sufficient to perform data analysis. From the end-user point of view, the data of different storage instances are represented in the local file system of the user's computer as a single mount point.

### 2.2. Simulations

Simulations are one of the important stages in modern experiments. They require a great deal of computing resources and produce a data volume that is comparable with the volume of raw data. Usually, simulations prove to be "data factories" similar to the experimental facilities. So, we propose to consider the simulations as a specific source of data (like experimental facilities). Thus, simulation data should be uploaded to the particular storage by a special service.

### 2.3. Analysis

The next pillar of the data life cycle is the data analysis itself. This task requires the delivery of requested data, access to computing facilities, and software for the analysis. There are two main approaches to data analysis in physics: conventional analysis and machine learning. In the first case, a user implements an algorithm inspired by the physical model of the phenomena under consideration. In the second case, one uses an artificial neural network technique with supervised or unsupervised learning. For the time being, this second technique, which has proven its efficiency for image recognition, is actively developed by different experiments equipped with telescopes, particularly by TAIGA for its Imaging Atmospheric Cherenkov Telescopes (IACT) [30].

### 2.4. Sharing and Open Access

The open access to the data is provided by the standard way under a specially formulated access policy. The policy depends on the local policy of integrated storage units (i.e., data owners).

---

7　https://www.ncbi.nlm.nih.gov/pubmed/26978244.
8　https://cernvm.cern.ch/portal/filesystem.

### 2.5. Education in Data Science

We will achieve this target by a using special service (i.e., web portal) based on the HUBzero platform. The platform will supply users with educational courses, documentation, and exercises on Monte Carlo simulation; examples of data analysis; introductions to the principle of metadata; and so on.

### 2.6. Archive

This target will be achieved by data provenance tracking. This tracking must store a full history of the data starting from the initial uploading. The history should include who processed the data and when, what kinds of software were used and their versions, what kind of calibration was used, etc. It should also provide a fast check of the data consistency. For example, the system should alarm if two chunks of data are processed under different calibration conditions. For this purpose, Merkle trees can be used. It is also possible to pack old data and upload it to offline storage (e.g., tapes). However, we do not suppose to fully solve this task here.

### 2.7. KCDC Extension

KCDC is an already-existing web portal where data of the KASCADE-Grande experiment are made available for the interested public (i.e., the methodological concept for this kind of data center is already developed). The web portal uses modern technologies, including standard internet access and interactive data selections. However, even if the primary target is the user community or "any interested scientist", both the data and the tools have to be refurbished in order to be usable without the detailed and highly specialized knowledge that is currently only available within internal collaborations.

The research plan in order to reach the project's goals in terms of providing Big Data science includes sophisticated methods and tools: the development of an adapted distributed system for Big Data analysis as well as its implementation at the large computing facilities in SINP MSU and SCC KIT. Then, the system needs to provide fast and reliable user access to the full dataset. A fast data exchange is foreseen to be reached via caching filesystems CVMFS and microservice technology using REST architecture. Further, the development of algorithms for the Big Data analysis of astrophysical experiments (particularly using machine learning) has to be pursued, and support of the soft- and hardware for the full data life cycle will be given using, for example, blockchain technology.

We propose to extend KCDC and even generalize cosmic ray data centers in order to preserve the intellectual value of the experiments and to further exploit their scientific contents beyond the lifetime of the instruments' operation. We think that a full return from the collaborations back to the society that funded this endeavor can best be achieved if the original data and the accompanying software tools are made publicly available in an open-access manner. The data center(s) will offer great and unique opportunities to people that would not be able to access such data otherwise, and will also provide a basis for education and outreach to the general public. This demand is at the heart of Big Data science. The amount of work needed to install and run such a web portal providing data from two independent experimental facilities with international collaborations should not be underestimated. Constant improvements of the availability and usability are needed. In addition, only a small part of the available data have been made available until now. Adding the remaining detector components will require processing of the raw data and updates to the documentation to cover the added observables. To enhance the usability of the data, the extensive set of simulations may also have to be added.

KCDC was implemented as a plugin-based framework to ensure an easy way to adjust, exchange, or remove components as needed. However, before the software can be released as open source, the coverage of the code by functional and unit tests has to be significantly improved. In addition, while there is a great deal of documentation on the published data available, there is almost no documentation on the usage of the software itself, or on the development of the software. Although it

has been kept intuitive, the extensive possibility to configure the web portal via an admin web interface makes such a detailed documentation necessary. Once published, user monitoring and feedback have to be taken into account to further improve the software. The possibility to include plugins and patches implemented by users has to be considered. Legal issues regarding ownership of the data have to be considered so as to avoid breaking the rules of the collaborations.

## 3. First Results

### 3.1. Metadata Architecture for Cosmic Ray Experiments

Since the amount of data is huge and their structures are diverse, a direct search within the data would be extremely slow and resource-consuming, and thus will not be implemented. Fortunately, the data have a common metadata format that includes time, place, atmospheric conditions, etc. A centralized database containing the metadata of all events from both experiments will be used to process data retrieval requests. The proposed database structure is presented in Figure 2[9]. If any request uses properties not included in this database, then the appropriate information must be extracted from the data and inserted into the metadata registry.

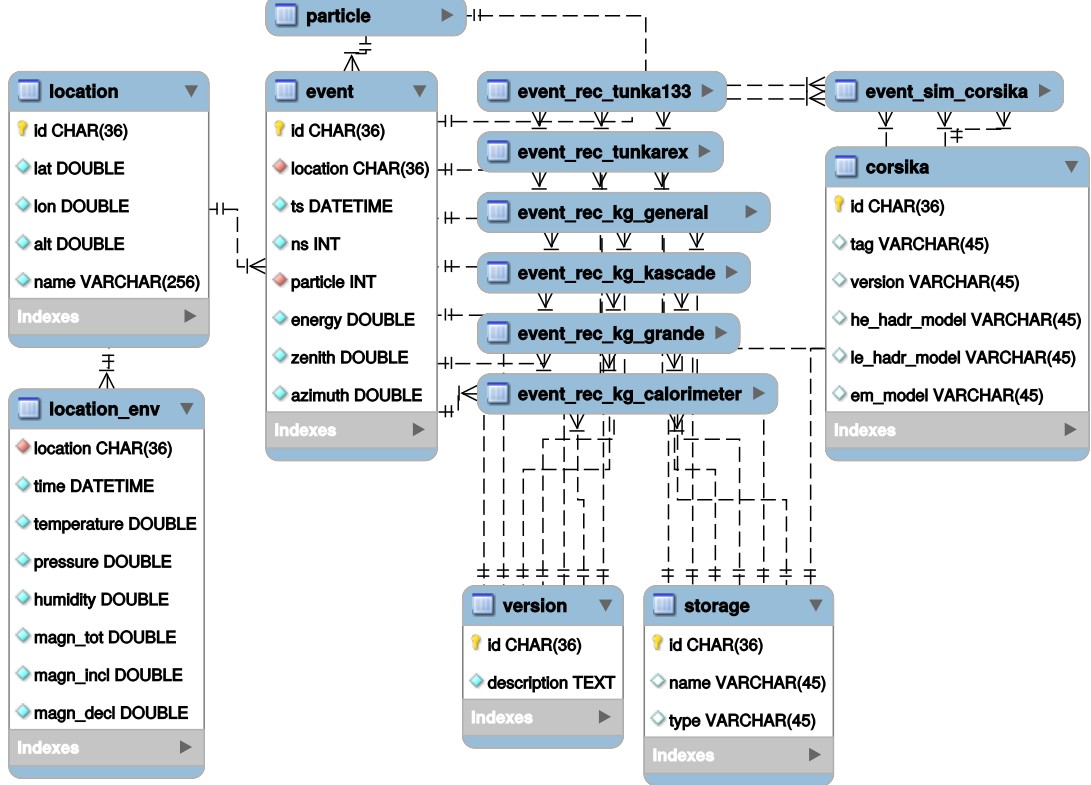

**Figure 2.** Proposed design of a database containing event metadata. The structure and plot were produced using MySQL Workbench software.

### 3.2. Data Analysis

We followed a machine learning approach to data analysis by the example of Monte Carlo simulation for the TAIGA-IACT imaging air Cherenkov telescope [31]. Our choice was to apply the convolutional neural network (CNN) technique to solve the problem of gamma ray identification. A CNN is a kind of artificial neural network that uses a special architecture that is particularly

---

9   https://www.mysql.com/products/workbench/.

well-adapted to classifying images. Today, CNNs are used in most neural networks for image recognition. However, only three CNN-related attempts have been made in IACT data analysis, all of them in the last two years: muon image identification for the VERITAS telescope [32], gamma-ray identification for the Monte Carlo simulation of a standalone telescope in the upcoming CTA project [33], and gamma-ray identification for the stereoscopic mode of the four H.E.S.S. telescopes [34].

In our CNN approach, we used Monte Carlo simulation for the TAIGA-IACT telescope. Datasets of gamma-ray images and hadron background (proton) images were simulated for the conditions of real observations (Figure 3). They were split into two parts for learning and testing, and various CNN versions were trained using two popular deep learning frameworks: PyTorch [35] and TensorFlow [36]. CNN performance estimation was a blinded study—a random proportion of test samples (blind samples) was used to estimate identification quality.

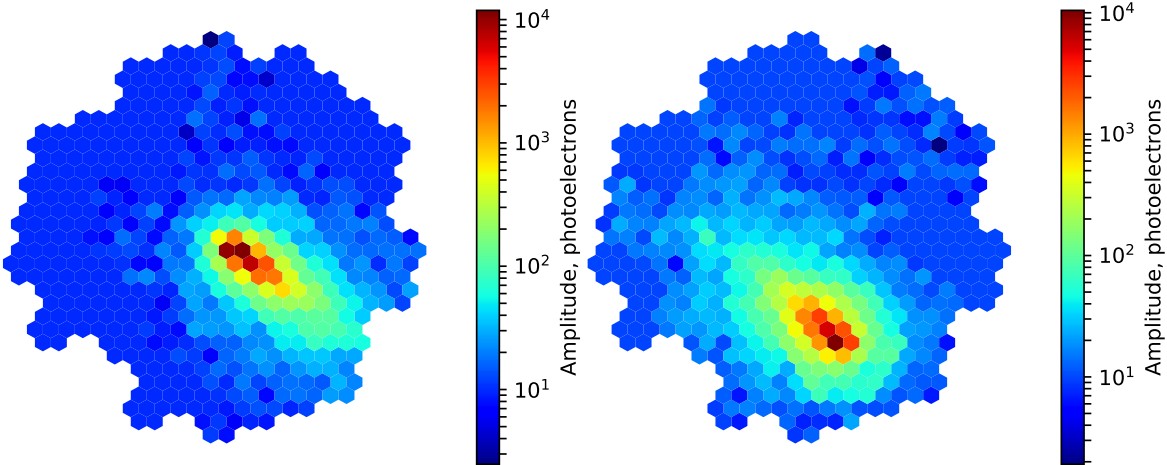

**Figure 3.** Examples of the Tunka Advanced Instrument for cosmic rays and Gamma Astronomy (TAIGA)-Imaging Atmospheric Cherenkov Telescopes (IACT) simulation images: gamma-ray (**left panel**) and background particle (proton, **right panel**). According to conventional gamma-ray identification techniques, image dimensions and orientations depend on the particle type.

The quality of identification allows the suppression of background (proton) events by a factor of 30 while keeping 55% of true gamma ray events. This is much better than the quality after a simple conventional analysis (a system of two consecutive cuts), which allows the suppression of proton events by only a factor of 8. After this technique is improved and experimentally verified, it will become part of the dedicated software for data analysis within the project.

### 3.3. Educational Resources

The HUBzero platform for the educational issues in astroparticle physics has been installed on the cloud infrastructure of the shared equipment center of the integrated information and computing network of the Irkutsk research and educational complex[10]. Currently, the educational resources are under development. They continue to be filled with documentation, educational courses, data, and tools for simulation and data analysis. The first experience of the application of this educational resource as a framework was received at the ISAPP-Baikal Summer School[11] "Exploring the Universe through multiple messengers". Due to lack of an Internet connection at the location of the Baikal Summer School, the platform was installed locally. This allowed the organizers of the school to spread the conference materials, lectures, and student reports on the site, so the participants had the opportunity to access the school materials online. Additionally, the participants could post their

---

[10]  http://net.icc.ru.
[11]  https://astronu.jinr.ru/school/current.

impressions via photos and video comments on the school's page. When all the activities within the school were completed, all resources were synchronized back with the online server. A screenshot of the web page with school materials can be found in Figure 4.

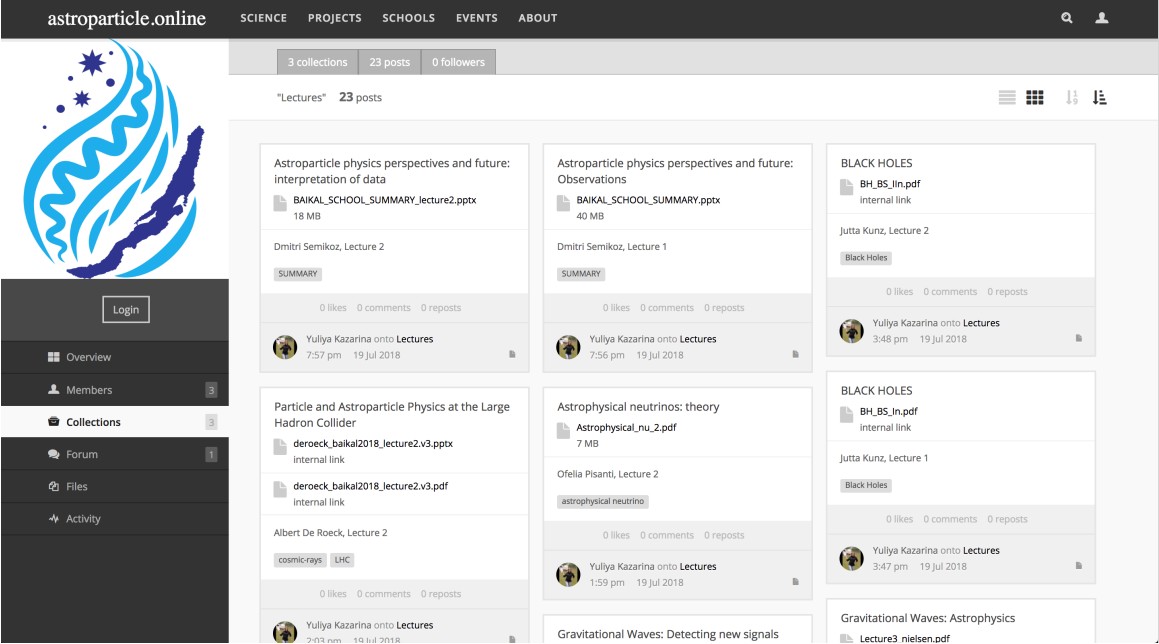

**Figure 4.** Screen shot of education materials of the ISAPP-Baikal Summer School held in 2018. It can be accessed via https://astroparticle.online/groups/bss.

### 3.4. Raw Binary Data Sharing and Reuse

One of the important issues to be considered is how to archive raw binary data to support their availability and efficient reuse in the future [37]. There are currently five binary file formats used in the various TAIGA projects. They provide a representation of raw data obtained from five TAIGA sub-facilities: the gamma ray setups TAIGA-HiSCORE and TAIGA-IACT [31] and the cosmic ray setups Tunka-133 [38], Tunka-Rex [39], and Tunka-Grande [40]. The long-term preservation of raw binary data as originally generated is essential for re-running analysis and reproducing research results in the future. In this case, the raw data need to be well-documented and accompanied by some readers (i.e., software for parsing these data). In addition, the format has to be compatible with the formats used in KASCADE-Grande.

Some of the state-of-the-art tools for formally describing binary data formats can provide a sufficient solution for the issues of documenting and parsing raw astroparticle physics data. We used two of them, Kaitai Struct[12] and FlexT[13], for formally describing TAIGA binary data formats, documenting, and parsing library generation. For example, Figure 5 demonstrates a diagram for the Tunka-133 file format specification presented in Kaitai Struct. As a result, we generated libraries for reading each format in target programming languages (including C/C++, Java, Go, JS, and Python). The libraries were successfully tested on real TAIGA data, which also helped us to identify the small portion of corrupted data and fix it.

---

[12] http://kaitai.io.
[13] http://hmelnov.icc.ru/FlexT.

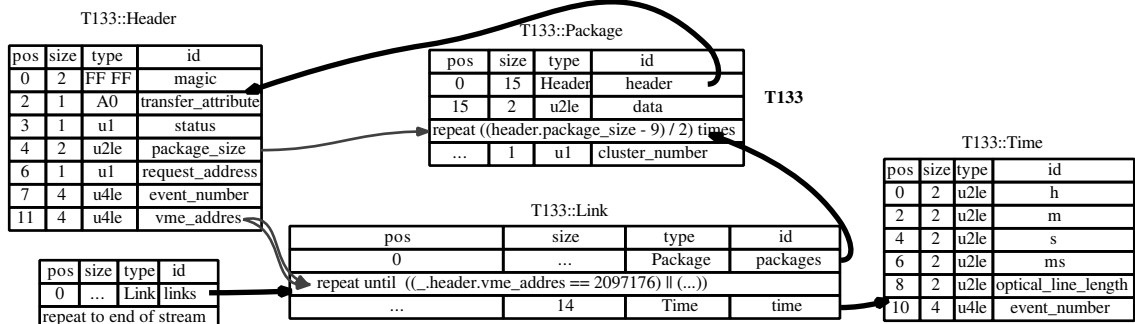

**Figure 5.** Tunka-133 file format specification presented in Kaitai Struct.

This result demonstrates a possible path for describing, sharing, and reusing binary file formats in astroparticle raw data. It can also be useful for other experiments, where raw binary data formats remain weakly documented or some parsing libraries for contemporary programming languages are required. We intend to use them to transfer raw data among web services that we are currently developing. They can also simplify the software development for data aggregation from various sources in the case of multi-messenger analysis.

## 4. Conclusions

The described pilot-scale initiative will have a significant long-term impact on the publication and release policies of present and future facilities in astroparticle physics and beyond. The resulting data centre and the experiences gained within this project will serve as a proof-of-principle that a public data centre opens the door to new methods of data analysis and to a new strategy of open science. In addition, it provides a concept for the required data release of forthcoming large-scale experiments in astroparticle physics, in particular as a dedicated facility spanning many different experiments.

This new strategic approach for astroparticle physics is possible because KCDC is already accepted by the community as a forerunner, but needs to be consolidated and matured in its scientific and technological performance to be ready for global use. The present initiative is a necessary step in this direction. In this sense, the results of this project will validate the concept of a widely usable public data centre in astroparticle physics.

We believe that our innovative approach will also be used in astroparticle physics beyond the present project. Plans are underway to expand the number of experiments by exporting data from other scientific collaborations. Our approach will rapidly advance the research into fundamental properties of matter and the Universe. It is noteworthy that the suggested approach can not only be used in the specified field, but can also be adapted to other scientific disciplines.

**Author Contributions:** Conceptualization, A.H. (Andreas Haungs), D.K. (Dmitriy Kostunin), and A.K.; Investigation, O.F., E.K., A.M., S.P., E.P., D.S., and D.Z.; Methodology, I.B., A.D., J.D., O.F., A.H. (Andreas Haungs), A.H. (Andreas Heiss), D.K. (Dmitriy Kostunin), Y.K., E.K., D.K. (Donghwa Kang), A.K., A.M., M.-D.N., S.P., E.P., A.S. (Alexey Shigarov), D.S., A.S. (Achim Streit), V.T., D.W., J.W., and D.Z.; Project administration, A.H. (Andreas Haungs) and A.K.; Resources, I.B., A.H. (Andreas Haungs), D.K. (Dmitriy Kostunin), Y.K., A.M., M.-D.N., S.P., A.S. (Alexey Shigarov), and D.S.; Software, A.D., J.D., A.H. (Andreas Heiss), D.K. (Dmitriy Kostunin), A.K., A.M., M.-D.N., S.P., D.S., A.S. (Achim Streit), V.T., D.W., J.W., and D.Z.; Supervision, A.H. (Andreas Haungs), D.K. (Dmitriy Kostunin), and A.K.; Validation, O.F., E.K., A.M., S.P., E.P., D.S., and D.Z.; Writing—original draft, A.H. (Andreas Haungs), Y.K., D.K. (Dmitriy Kostunin), A.K., E.P., and A.S. (Alexey Shigarov); Writing—review & editing, A.H. (Andreas Haungs), D.K. (Dmitriy Kostunin), A.K., M.-D.N., and E.P.

**Funding:** This work was financially supported by Russian Science Foundation Grant 18-41-06003 (Sections 2 and 3) and Helmholtz Society Grant HRSF-0027.

**Acknowledgments:** The developed educational resources were freely deployed on the cloud infrastructure of the Shared Equipment Center of Integrated Information and Computing Network for Irkutsk Research and Educational Complex (http://net.icc.ru).

**Conflicts of Interest:** The authors declare no conflict of interest.

## Abbreviations

The following abbreviations are used in this manuscript:

KASCADE    KArlsruhe Shower Core and Array DEtector
TAIGA      Tunka Advanced Instrument for cosmic rays and Gamma Astronomy
IACT       Imaging Atmospheric Cherenkov Telescope
HiSCORE    High-Sensitivity Cosmic ORigin Explorer
KCDC       KASCADE Cosmic-ray Data Centre
SCC KIT    Steinbuch Centre for Computing Karlsruhe Institute of Technology
SINP MSU   Skobeltsyn Institute of Nuclear Physics Lomonosov Moscow State University

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
