# Peer review of "Russian–German Astroparticle Data Life Cycle Initiative"

_data, 2018_

Round 1

Reviewer 1 Report

Reviewer comments:

Journal Data (ISSN 2306-5729)

Manuscript ID data-379348

Title: “Russian-German Astroparticle Data Life Cycle Initiative”

Authors: Igor Bychkov , et al.

In this paper authors report current work and stage of the Rus-Ger Astroparticle Data Life Cycle Initiative and service called ASTROPARTICLE.ONLINE. The manuscript represents interesting contribution with mainly preliminary results but with amount of new information. I believe that the results will be interesting for potential readers and I would like to recommend publications of this paper. 

Before publication I ask authors to correct following:

-The abstract should be rewritten, i.e. shortened and important stressed.

-  Authors should give full affiliation e.g.

1 Matrosov Institute for System Dynamics and Control Theory, Siberian Branch of Russian Academy of Sciences, Irkutsk, Russia => 1 Matrosov Institute for System Dynamics and Control Theory, Siberian Branch of Russian Academy of Sciences, Irkutsk,  P. O. Box 1233, 664033, Irkutsk, Lermontova st., 134

same goes for all other affiliations.

-In the first paragraph in the Introduction (page 1), the authors should insert appropriate references because currently there is not one in this paragraph. Also, authors could insert more references from the literature in the rest of the manuscript.

-Page 3, line 85 : “The other example is the ground-based experiment LSST [6], providing over 3 gigapixels per image ...” => The other example is the ground-based experiment LSST [6], which will provide over 3 gigapixels per image ...”     

LSST  first light is expected around next year or later.

-In Author Contribution authors should use initials e.g.

Author Contributions: Conceptualization, Andreas Haungs, Dmitriy Kostunin and Alexander Kryukov; … => Author Contributions: Conceptualization, A. H., D. K. and A. K.; …

-In References: All references should be written in the same way, according to the journal style.

For Journal Name authors should use Abbreviated Journal Name

e.g. in

ref 12 Physics of Atomic Nuclei => Phys Atom Nucl

ref 7 Industrial and Corporate Change => Indus. & Corp. Change

also in

ref 1 Haungs, A.; others. => Haungs, A.; et al.

ref 6 Abell, P.A.; others. => Abell, P.A. et al.

…etc

Respectfully,

Author Response

1. The abstract was rewritten to a shorter version; important statements are stressed.
2. We added the full affiliations of authors with postal addresses.
3. The Introduction section was extended, appropriate references were added.
4. The sentence "The other example is the ground-based experiment LSST, providing over 3 gigapixels per image …" (page 3, line 85) was corrected as the reviewer 1 mentioned.
5. We replaced the author names with initials in the Author Contribution section.
6. All references were corrected according to the journal style. Now they contain abbreviated journal names.

Reviewer 2 Report

This manuscript does not present original research. It is a proposal or perspective (or whitepaper) paper for a service called ASTROPARTICLE ONLINE. Currently, there are only 3 types of manuscripts in this journal: Data Descriptors, Articles, and Review. The editor should decide which type should be assigned to it: perspective or whitepaper article. 

Author Response

We are grateful to the reviewer 2 for the given remark. We agree that our article goes deep into the perspective of the presented initiative. Nevertheless, it presents our current contribution to the progress of the initiative. We added a list of the results presented in the paper (see the end of the Introduction section). We hope this clarifies the reason why we think that our contribution can be considered as an article too.

Reviewer 3 Report

The manuscript "Russian-German Astroparticle Data Life Cycle Initiative" by I. Bychkov et al   proposes the creation of  an 'open science system which enables to publish, store, search, select and analyse astroparticle physics data". If this was a proposal and not a paper then it would have to be reviewed on a different basis judging its feasibility, scope, etc. As a paper, however, I do not find something very  wrong with it except from the fact that is putting more emphasis on the management part and less on the science. Also it contains some parts that need more justification. For example, I found the statement  "charged particles, gamma-rays or neutrinos, measured by different large-scale facilities globally distributed, should be combined to obtain increased knowledge of the high-energy Universe" confusing. While I can understand that the simultaneous detection of gamma-rays and neutrinos can help us deduce the physics of sources like AGNs (see the recent IceCube discovery of neutrinos coming from the direction of blazar TXS 0506+056), I cannot understand what would be the role of Cosmic Rays in this picture. Furthermore, gamma-rays by themselves do not tell the whole story but there are other wavebands that have to be combined in order to understand the physics of High Energy sources as at least 25 years of multiwavelength observations have shown.  Therefore, I think that the authors in order to make their arguments more convincing they will have to put more emphasis on the physical arguments of their manuscript and answer the question on what will be the scientific gains coming out of this effort. As it is now. the manuscript reads more like a dry technical report.

Author Response

We put more emphasis on the physical arguments in the revised paper. In particular, we clarified the role of Cosmic Rays, answering the question of the reviewer 3 in the Introduction section. While the neutrinos and gamma rays point directly to the source, the cosmic rays are heavily declined in the galactic and extra-galactic magnetic fields. Nevertheless, their anisotropy can indicate the nearest sources as well as point to the possible distant sources of ultra-high-energy cosmic rays. Moreover, with increasing the energy of cosmic rays the impact of magnetic fields decreases, which allows one to perform extremely high energy proton astronomy.
